# Giant dielectric tunability in ferroelectric ceramics with ultralow loss by ion substitution design

Ruitao Li[1], Diming Xu[1] ✉, Chao Du[1], Qianqian Ma[2], Feng Zhang[3], Xu Liang[2] ✉, Dawei Wang[3] ✉, Zhongqi Shi[3], Wenfeng Liu[4] & Di Zhou[1] ✉

Due to their responsiveness to modulation by external direct current fields, dielectric tunable materials are extensively utilized in integrated components, such as ferroelectric phase shifters. Barium strontium titanate ceramics have been considered the most potential tunable materials for a long time. However, the significant dielectric loss and high voltage drive have limited their further applications. Recently, $Bi_6Ti_5WO_{22}$ ceramic has regained attention for its high dielectric tunability with low loss. In this study, we judiciously introduce $Nb^{5+}$ with a larger ionic radius, replacing $Ti^{4+}$ and $W^{6+}$. This successful substitution enables the modulation of the phase transition temperature of $Bi_6Ti_5WO_{22}$ ceramics to room temperature, resulting in superior tunable properties. Specifically, the $0.7Bi_6Ti_5WO_{22}–0.3Bi_6Ti_4Nb_2O_{22}$ ceramics exhibit giant tunability (~75.6%) with ultralow loss (<0.002) under a low electric field (1.5 kV/mm). This tunability is twice that of barium strontium titanate ceramics with a similar dielectric constant and only one-tenth of the loss. Neutron powder diffraction and transmission-electron-microscopy illustrate the nanodomains and microstrains influenced by ion substitution. Density functional theory simulation calculations reveal the contribution of ion substitution to polarization. The research provides an ideal substitute for tunable material and a general strategy for adjusting phase transition temperature to improve dielectric properties.

Capacitors constitute the fundamental components of microwave-integrated devices within electronic systems, and their performance is primarily governed by the dielectric materials employed. Dielectric tunable materials are materials whose dielectric constant can be modulated by an external direct current field (E). Due to their widespread application in devices such as microwave phase shifters and phased-array radars, dielectric tunable materials have attracted extensive research interest[1–6]. A standard parameter for evaluating the tunable performance of materials is dielectric tunability (T),

which represents the maximum nonlinearity of reversible dielectric constants[1]:

$$T = \frac{\varepsilon_{r(0)} - \varepsilon_{r(E)}}{\varepsilon_{r(0)}} \times 100\% \tag{1}$$

where $\varepsilon_{r(0)}$ is the dielectric constant of the material without external direct current bias, and $\varepsilon_{r(E)}$ is the dielectric constant under an applied direct current bias electric field (E).

[1]Electronic Materials Research Laboratory & Multifunctional Materials and Structures, Key Laboratory of the Ministry of Education & International Center for Dielectric Research, School of Electronic Science and Engineering, Xi'an Jiaotong University, Xi'an 710049, China. [2]State Key Laboratory for Strength and Vibration of Mechanical Structures, Xi'an Jiaotong University, Xi'an 710049, China. [3]School of Microelectronics & State Key Laboratory for Mechanical Behavior of Materials, Xi'an Jiaotong University, Xi'an 710049, China. [4]State Key Laboratory of Electrical Insulation and Power Equipment, Xi'an Jiaotong University, Xi'an 710049, China. ✉e-mail: diming.xu@xjtu.edu.cn; xliang226@xjtu.edu.cn; Dawei.wang@xjtu.edu.cn; zhoudi1220@gmail.com

In general, the dielectric tunability of materials is positively correlated with the strength of the applied electric field. In current research, the pursuit of higher $T$ often overlooks the applied electric field intensity. Methods such as manufacturing thin films are employed to significantly increase the intrinsic breakdown strength of materials, achieving higher tunability by applying high electric fields. However, this approach contradicts the practical requirement for lower bias voltage in real applications. Addressing this situation, Hu et al. introduced the concept of effective tunability $T_e$[7]:

$$T_e = \frac{T}{E} \tag{2}$$

where $T$ is the dielectric tunability at a specific applied electric bias, and $E$ is a specific applied electric bias ($E \geq 0.5$ kV/mm).

Barium strontium titanate ceramic ((Ba, Sr)TiO$_3$, abbreviated as BST) has been considered the most potential tunable material for a long time. In the BST solid solution series, Ba$_{0.65}$Sr$_{0.35}$TiO$_3$ (abbreviated as BST65) exhibits the highest effective tunability at room temperature, ranging from 0.15 to 0.4[7–10]. Such effective tunability meets practical requirements, but the large dielectric constant and high loss limit its further applications. The incorporation of organic compounds can reduce the dielectric constant of the material[7,11–16]. The introduction of the ferroelectric phase can reduce the dielectric loss of BST ceramics[17–21]. However, both methods inevitably lead to a decrease in the effective tunability of the material. Despite extensive research in these two directions, breakthrough progress has yet to be achieved. Over time, many new tunable ceramic systems have been discovered and studied. These include ferroelectric materials such as Pb(Zr, Ti)O$_3$, (Bi, Na)TiO$_3$, (Na, K)NbO$_3$, Ag(Ta, Nb)O$_3$, as well as some non-ferroelectric materials like (Bi, Zn, Nb)O$_7$[22–30]. However, these materials have not been able to successfully replace the BST ceramic system in the tunable field because of their inferior effective tunability compared to BST65. In summary, the insufficient effective tunability and high loss impede further research on tunable materials, thereby restricting the potential applications of both tunable materials and devices.

## Results

Bi$_6$Ti$_5$WO$_{22}$ ceramic (abbreviated as BTW) has recently regained attention for its high dielectric tunability with low loss in our previous work[31,32]. We detailed the preparation and structural characterization of dense and phase-pure BTW ceramic, showcasing a remarkable 40% tunability at 3 kV/mm[32]. The utilization of ion substitution to achieve solid solution ceramics has proven to be an effective strategy for enhancing dielectric performance[33]. Experimental evidence suggests that optimal effective tunability is typically achieved near the intermediate temperature (abbreviated as $T_m$), which is the reason why BST65 with a $T_m$ of approximately 20 °C has the highest tunability among solid solution ceramics. Since $T_m$ of BTW is approximately 10 °C, achieving better tunable performance requires shifting its $T_m$ toward higher values. Based on our previous studies on the structure of pure BTW, the main source of material polarization is the relative displacement of the Ti-O octahedra at the two Ti$^{4+}$/W$^{6+}$ shared sites[32]. Therefore, we believe that ion substitution at these positions can effectively tune the material's properties. Considering that the radius of Ti$^{4+}$ ions is 0.605 Å and that of W$^{6+}$ ions is 0.60 Å, we consider Nb$^{5+}$ ions with a radius of 0.64 Å to be a promising choice[34]. On the one hand, the relatively close ion radius allows Nb$^{5+}$ ions to bind to the existing lattice framework to form a solid solution without generating a second phase; on the other hand, the slightly larger ion radius enables the introduction of Nb$^{5+}$ ions to induce a change in lattice constant, thereby adjusting the phase transition temperature and affecting the dielectric properties. Additionally, the pentavalent state of Nb$^{5+}$ ions ensures that the structural charge balance is maintained when two Nb$^{5+}$ ions replace

one W$^{6+}$ and one Ti$^{4+}$ ion. Building upon this idea, we judiciously designed a series of $(1-x)$Bi$_6$Ti$_5$WO$_{22}$–$x$Bi$_6$Ti$_4$Nb$_2$O$_{22}$ ($0 \leq x \leq 0.6$) solid solution ceramics (abbreviated as BTW-BTN).

Figure 1d illustrates the X-ray diffraction (XRD) patterns of the solid solution ceramics. It can be observed that for $x \leq 0.5$, the XRD diffraction peaks match the characteristic $pnn$2 peaks of BTW. No distinct secondary phase peaks are evident, even for the composition until $x = 0.5$. However, a considerable presence of the secondary phase, Bi$_6$Ti$_3$WO$_{18}$, is observed for $x = 0.6$. This indicates that the maximum solid solubility of BTW-BTN is approximately 50%, a fact corroborated by scanning electron microscope (SEM) micrographs (Fig. 1e). With the Nb content increases, the grain size of the samples progressively increases, which is caused by the slight increase in sintering temperature with the introduction of Nb. However, up to x = 0.4, the surface grains of the samples are very uniform, with no presence of any secondary phase. Thin-plate-shaped secondary phases become noticeable starting from $x = 0.5$ and are even more pronounced at $x = 0.6$. Energy dispersive X-ray spectrometry (EDS) analysis (Supplementary Fig. 1) confirms that these plate-like secondary phases are enriched in Bi. Combining SEM micrographs, EDS elemental analysis, and characteristic peaks in XRD, we preliminarily infer that the impurity phases are probable to be Bi$_6$Ti$_3$WO$_{18}$ and Bi$_3$TiNbO$_9$, which are considered to be iso-structural[35]. Moreover, the refined results of the room-temperature XRD patterns indicate that, with the increasing Nb content, the main XRD peak shifts to lower angles, indicating an enlargement of lattice constants due to the successful introduction of Nb (as depicted in Fig. 1d), with detailed XRD refinement results are shown in Supplementary Fig. 2. The theoretical density of the solid solution material was computed based on the refinement results. Through comparison with experimentally measured density, it was established that the relative densities of ceramics with compositions ranging from $x = 0.1$ to 0.4 consistently exceed 97% (Supplementary Fig. 3).

To comprehensively understand the impact of Nb incorporation on the dielectric performance of BTW-BTN, the temperature-dependent dielectric spectra were measured (Fig. 2a). With the increasing Nb$^{5+}$ content, the phase transition temperature of BTW-BTN shifts towards higher temperatures, leading to the successful generation of a series of solid solution ceramics with continuous variations in phase transition temperatures. For compositions with $x = 0.2$, 0.3, and 0.4, the peak dielectric constants experience a remarkable increase, reaching levels as high as ~11000. Supplementary Fig. 4 presents detailed temperature-dependent dielectric spectra for each component. The alteration of dielectric loss adheres to the general trend observed in composite ferroelectric ceramics: a peak near $T_m$ is followed by a decrease in loss beyond $T_m$, maintaining them at very low levels. Figure 2b depicts the relationship between $T_m$ and the content of Nb. The conductivity exhibits a similar pattern in its temperature-dependent spectra, showing a distinct transition near $T_m$ (Supplementary Fig. 5). Notably, $T_m$ exhibits almost linear variation with the introduction of Nb. Specifically, for compositions with $x = 0.3$ and 0.4, the phase transition temperatures are around 15 °C and 25 °C, respectively, close to room temperature. This suggests that the high dielectric constants and strong polarization in the vicinity of $T_m$ can be effectively harnessed, leading to elevated tunability.

The relaxation behaviors of BTW-BTN ceramics with different Nb$^{5+}$ contents exhibit some variations across frequencies. The activation energy for dielectric relaxation loss in BTW-BTN ceramics was calculated, and the relationship between ln $f$ and $1000/T$ for BTW-BTN ceramics with various Nb$^{5+}$ contents is presented in Fig. 2c. The activation energy can be determined from the slope using the following formula[36]:

$$ln f = ln f_0 + \frac{-E_a}{k_B}\frac{1}{T} \tag{3}$$

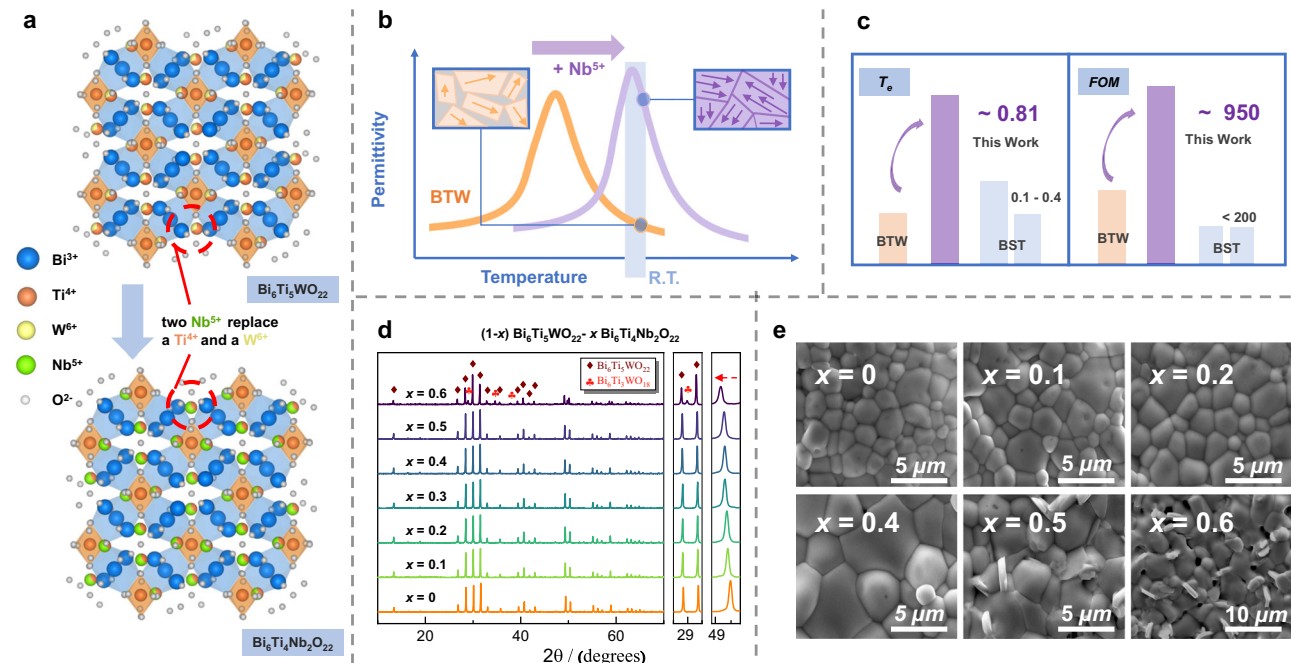

**Fig. 1 | Design and identification of (1-$x$) Bi$_6$Ti$_5$WO$_{22}$−$x$ Bi$_6$Ti$_4$Nb$_2$O$_{22}$ (0 ≤ $x$ ≤ 0.6) solid solution ceramics. a** Preparation route of solid solution ceramics based on ion substitution strategy. **b** Expectations for the introduction of Nb$^{5+}$. **c** Enhancement of dielectric tunability and comparison with common materials by ion substitution design. **d** X-ray diffraction (XRD) patterns of the ceramics. **e** Scanning electron microscope (SEM) image of the ceramics surface.

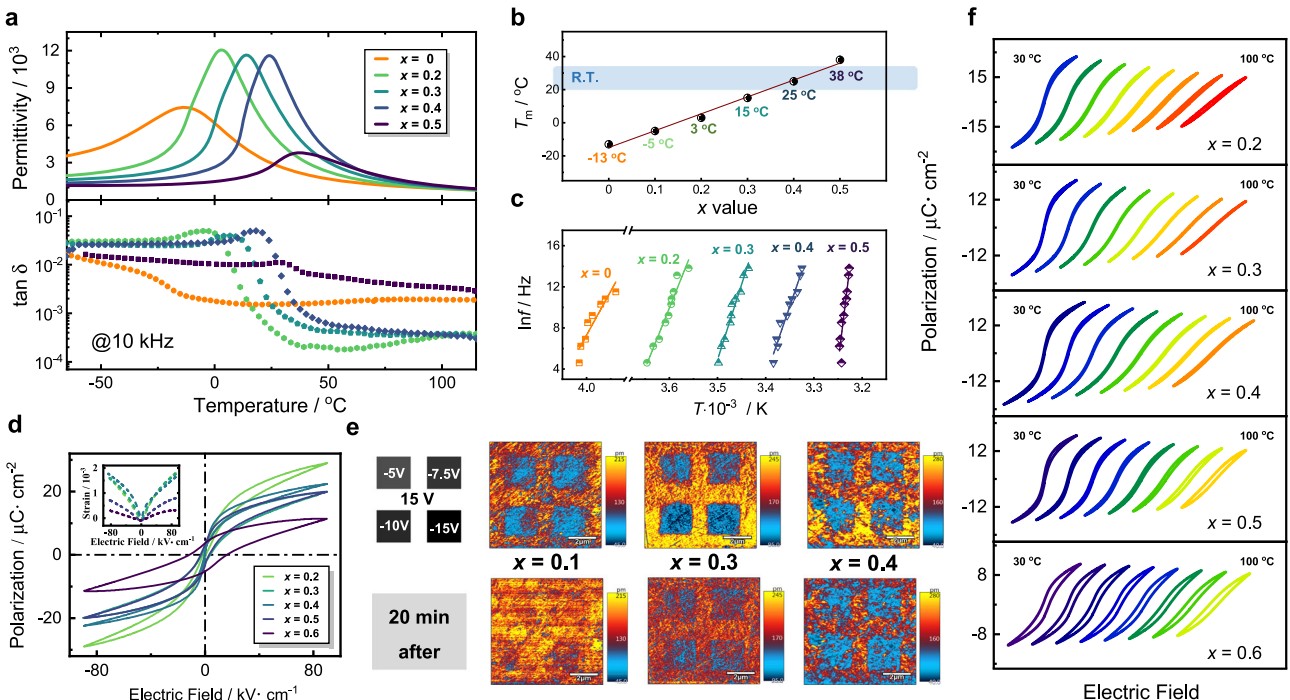

**Fig. 2 | Comparison of ferroelectric and dielectric properties of (1-$x$) Bi$_6$Ti$_5$WO$_{22}$−$x$Bi$_6$Ti$_4$Nb$_2$O$_{22}$ (0 ≤ $x$ ≤ 0.6) solid solution ceramics. a** Temperature-dependent dielectric spectra. **b** Variation curve of $T_m$ under different Nb$^{5+}$ contents. **c** Relaxation time as a function of 1000/$T$. **d** Response of polarization and strain to an electric field of 60 kV·cm$^{-1}$ and 10 Hz. **e** Out-of-plane atomic force microscopy (AFM) amplitude images after poling treatment with different electrical voltages and relaxation durations. **f** The polarization-electric field hysteresis loops of ceramics under different temperatures at the electric field of 100 kV·cm$^{-1}$.

where $f$ is the measured frequency value, $T$ represents the inflection point temperature of loss for each composition at different frequencies, and $k_B$ is the Boltzmann constant. Although this formula yields activation energies with some error, it qualitatively allows for comparing relaxation energy strengths among different compositions.

From the graph, it can be observed that the slope increases with higher Nb$^{5+}$ content. This behavior arises due to the enlargement of grain size upon Nb$^{5+}$ introduction, which results in an increase in both grain boundary charges and space charges within the ceramic. Under the influence of an external electric field, the transfer and reorientation of

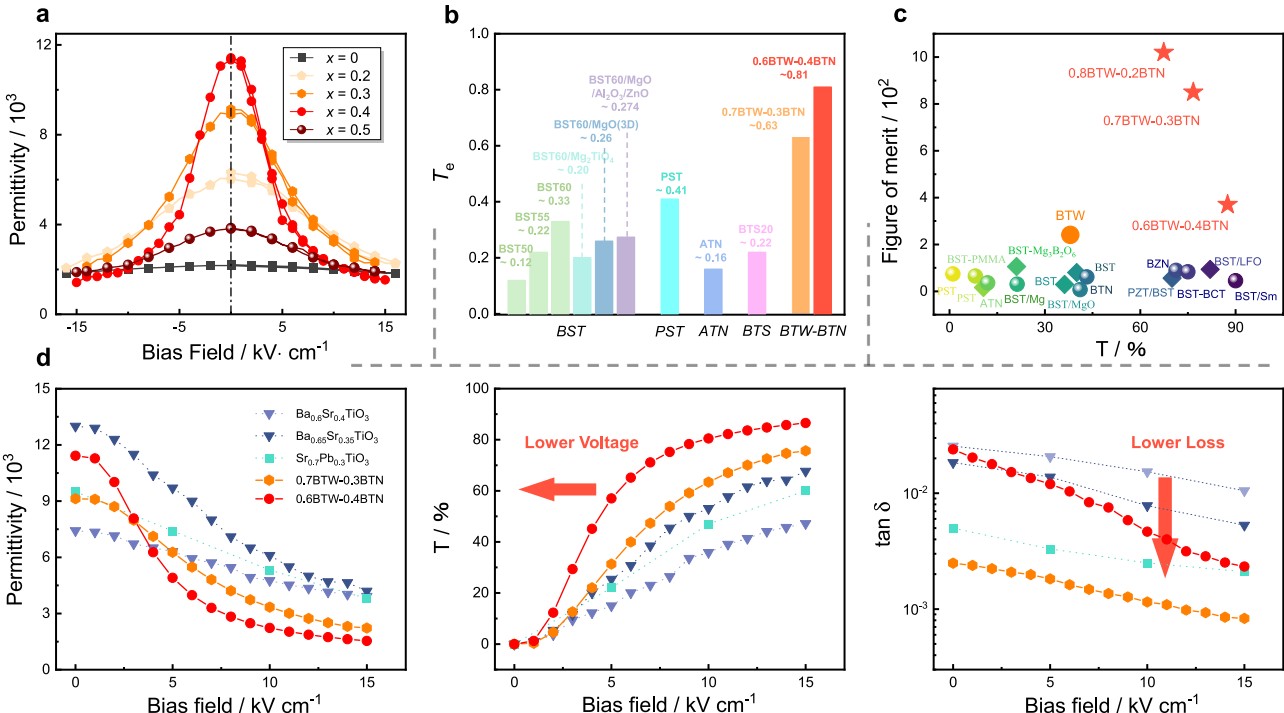

**Fig. 3 | Excellent dielectric tunable performance of $(1-x)Bi_6Ti_5WO_{22}-xBi_6Ti_4Nb_2O_{22}$ $(0 \leq x \leq 0.6)$ solid solution ceramics. a** Reversible-nonlinear dielectric spectra with the electric field. **b, c** Comparison of the effective dielectric tunability and the figure of merit with selected materials[6,9,14,28]. **d** Comparisons of comprehensive properties (permittivity, tunability, and dielectric loss) with a direct current electric field between our study and other representative tunable ceramics with similar permittivity.

free charges become more challenging, consequently increasing activation energy. This enhanced energy can be partly characterized through atomic force microscopy (AFM) imaging. Figure 2e illustrates AFM images of the induced compressive stress for different compositions at room temperature. Supplementary Fig. 6 displays the surface morphology of the three samples, which appear sufficiently smooth. The upper images show amplitude images captured immediately after applying a reverse voltage to a portion of the sample surface, while the lower images are captured from the same location after 20 minutes. The disparity between the two sets of images reflects the duration of the polarization process. Notably, in the case of the $x = 0.1$ composition, there is almost no amplitude remaining in the image taken after 20 minutes, whereas for the $x = 0.4$ composition, the difference between the images captured after 20 minutes and immediately after applying the voltage is minimal. The behavior of the $x = 0.3$ composition falls between these two extremes. The situation of phase behavior shows a similar trend (Supplementary Fig. 7). This indicates that with the introduction of $Nb^{5+}$, the polarization of the sample at room temperature becomes more challenging, resulting in poorer relaxation behavior. In ferroelectric materials, such behavior is often attributed to changes in domain structures, which will be discussed in detail in the subsequent sections.

The measurement of ferroelectric properties was also successfully conducted. Figure 2d illustrates the polarization-electric field hysteresis loops (P-E loops) and strain coefficients for different solid solution ceramics. The P-E loops of $x = 0.3$ and $x = 0.5$ overlap, making them difficult to distinguish. The curves for each component separately plotted can be found in Supplementary Fig. 12. As $Nb^{5+}$ content increases, the polarization at the same field strength weakens, and the P-E loops exhibit more characteristic ferroelectric behavior. This phenomenon can be easily understood as the $T_m$ shifts towards room temperature. Notably, BTW-BTN exhibits significant uniaxial strain, with considerable strain (0.1% to 0.2%) even at low electric fields of 80 kV/cm. Compositions $x = 0.2$ and $x = 0.3$ both demonstrate strain

strengths of nearly 0.2%, which is quite remarkable for lead-free perovskite ceramics. The curves of different components under breakdown field strength are shown in Supplementary Fig. 8. Figure 2f compiles temperature-dependent P-E loops for different solid solution compositions. Each composition was tested at intervals of 10 °C from 30 °C to 100 °C, resulting in eight P-E loops. Different colors represent the degree of curve in the loops, indicative of the strength of the ferroelectric behavior (red lines are more linear, while blue lines are more curved). This set of temperature-dependent P-E loops reveals a strong regularity, as the $T_m$ shifts towards higher temperatures with increasing $Nb^{5+}$ content. Moreover, the temperature at which the P-E loops tend towards linearity also shifts towards higher temperatures, demonstrating a close alignment with the dielectric performance.

Figure 3a illustrates the dielectric tunability performance of BTW-BTN ceramics measured at room temperature under 10 kHz. The response of the dielectric constant and dielectric loss of different components to the electric field is shown in Supplementary Fig. 9. It is evident that the introduction of $Nb^{5+}$ induces a significant leap of tunability compared to the original BTW ceramics. The most pronounced improvement is in compositions $x = 0.3$ and $x = 0.4$. However, the tunability property diminishes for the $x = 0.5$ sample as room temperature is below its $T_m$. The concept of effective tunability $T_e$ has been introduced in the introduction. When the applied electric field intensity is 1 kV/mm, the calculated $T_e$ of the $x = 0.3$ compositions is approximately 0.634 mm/kV. That value of the $x = 0.4$ compositions reaches an impressive 0.805 mm/kV (although with relatively higher loss), significantly exceeding the $T_e$ reported for existing material systems. Compared to common tunable materials, the improvement in the $T_e$ of BTW-BTN is intuitively illustrated in Fig.3b. (The comparative data are all calculated also calculated under the electric field of 1 kV/mm.) Moreover, the figure of merit (FOM) is also a common parameter used to evaluate the quality of dielectric tunable materials and is defined as the ratio of tunability to dielectric loss. Figure 3c compares the FOM values of common tunable materials with those of the BTW-

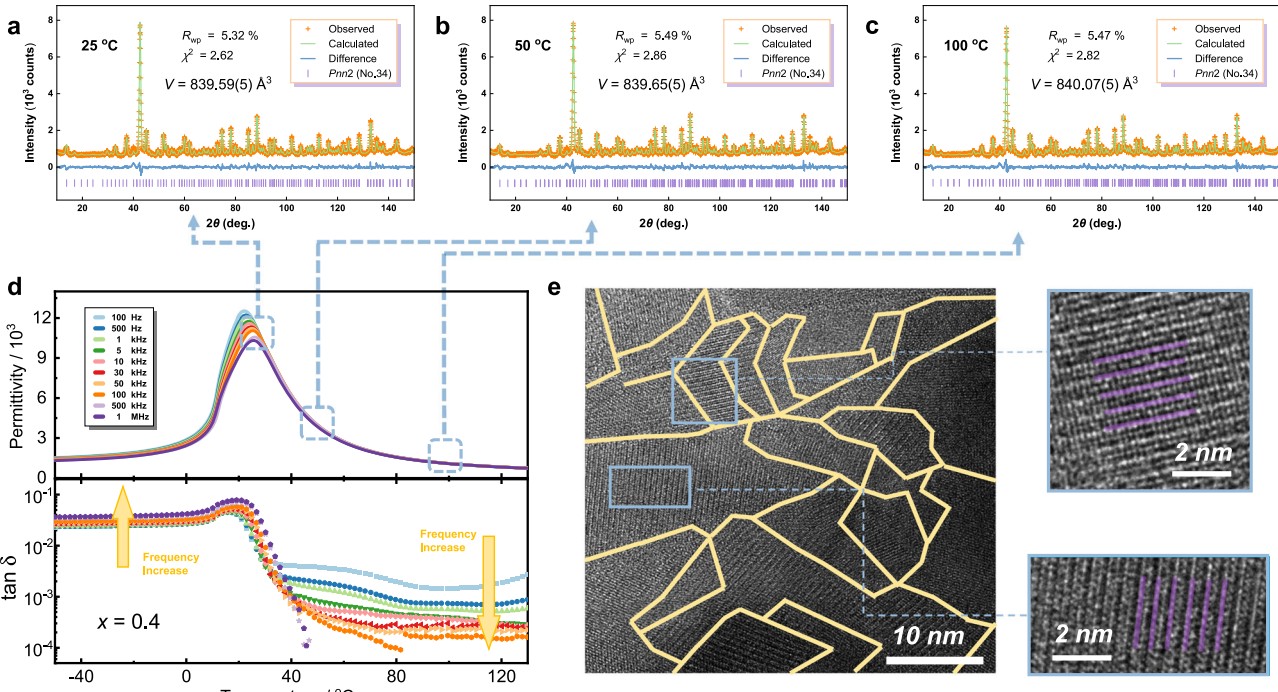

**Fig. 4 | Phase and domain micro-structures of $0.6Bi_6Ti_5WO_{22}$–$0.4Bi_6Ti_4Nb_2O_{22}$ and their effects on properties. a–c** Combined Rietveld refinement plots of constant wavelength neutron powder diffraction data using the *Pnn*2 model at three different temperatures. **d** Temperature-dependent dielectric spectra with several temperature points marked for neutron powder diffraction refinement. **e** High-resolution transmission electron microscopy (HRTEM) image with yellow lines delineates the nanodomains. The magnified image of the selected area shows clear domains with purple lines for the strips.

BTN ceramics. It can be seen that due to its extremely low loss of $10^{-3}$, BTW-BTN exhibits a much higher FOM value than common materials.

Figure 3d shows the comparison between our study with $Ba_{0.65}Sr_{0.35}TiO_3$ (BST65), $Ba_{0.6}Sr_{0.4}TiO_3$ (BST60), and $Pb_{0.3}Sr_{0.7}TiO_3$ (PST30), which are the most representative tunable material with similar permittivity to our study[37,38]. Under the premise of similar dielectric constants, the dielectric loss of BTW-BTN is lower, and the bias voltage required to achieve the same tunability is lower. For example, if applied as an adjustable capacitor, to reduce the capacitance values of five materials of the same size and thickness by half, the applied bias electric fields are ~4.3 kV/cm($x = 0.4$), ~7 kV/cm($x = 0.3$), ~9 kV/cm(BST65), ~12 kV/cm(PST30), ~17 kV/cm(BST60), respectively. The fabrication of practical phase shifters involves various complex factors, and different devices have specific impedance-matching requirements and maximum dielectric constant requirements[5,39,40]. Due to the relatively high relative dielectric constant of BTW-BTN materials, further comprehensive research is needed for their practical applications in phase shifters. However, compared to the extensively researched BST60 ceramic, where the maximum dielectric constant is as high as 7200, BTW-BTN ceramics, particularly the 0.7BTW-0.3BTN composition, offer similar dielectric constants with lower dielectric loss and a relative tunability three times that of BST60. Modifications to reduce dielectric constant in BST systems, such as organic-inorganic composites and spark plasma sintering of thin films, have been well developed[37,41–49]. It can be foreseen that similar methods can be applied to 0.7BTW-0.3BTN ceramics to achieve effective tunability surpassing that of BST-based ceramics while reducing the dielectric constant.

To explore the origin of the significant tunability in BTW-BTN ceramics, neutron powder diffraction (NPD) Rietveld refinement and high-resolution transmission electron microscopy (HRTEM) observations were performed on the composition with the highest tunability, $0.6Bi_6Ti_5WO_{22}$–$0.4Bi_6Ti_4Nb_2O_{22}$. The results are shown in Fig. 4. Figure 4a–c displays the refined NPD patterns at three different temperatures (25 °C, 50 °C, and 100 °C) corresponding to different

stages in the dielectric constant curves in Fig. 4d. Detailed crystal information and refinement results are listed in Supplementary Table 1 to 4. The refined patterns at various temperatures illustrate $0.6Bi_6Ti_5WO_{22}$–$0.4Bi_6Ti_4Nb_2O_{22}$ trends to normal thermal expansion above 25 °C. Unit cell parameters increase to a certain extent with temperature, maintaining an orthogonal phase structure of $a \approx c > b$. The differences in the unit cell parameters between 25 °C and 200 °C are less than 0.01 Å. HRTEM images at room temperature (Fig. 4e) reveal denser and clearer nanoscale polar regions and twin boundaries than those observed in pure BTW (the comparison is shown in Supplementary Fig. 10). Enlarged images display distinctive alternating bright and dark stripes (highlighted in purple) with an average width of approximately 5.2 Å. This nanoscale domain structure may result in higher grain boundary charges and energy, thereby enhancing the dielectric constant and reducing relaxation behavior. Simultaneously, the control exerted by a direct current voltage on this reversible polarization behavior readily induces the rearrangement of these domain structures, leading to significant dielectric tunability.

DFT simulation calculations revealed the contribution of asymmetric structures and ion substitution to polarization (Fig. 5). Previous reports have shown that both $TiO_6$ octahedron, $WO_6$ octahedron, and $BiO_{12}$ polyhedron contribute to overall polarization in the *pnn*2 structure[32]. This work quantitatively presents the polarization enhancement brought by the orthogonal phase structure. Compared with the BTW (34) model, which replaces two-thirds of $Ti^{4+}$ with $W^{6+}$ on the cubic phase $BiTiO_3$ base oil in space group 34, BTW (3) model, which uses orthogonal phase $Bi_6Ti_5WO_{22}$, the calculated total polarization increases by approximately~11.2 $\mu C/cm^2$. The BTN (3) model (see Supplementary Fig. 11), based on BTW (3), replaces substitution.

## Discussion

A series of solid solution ceramics $(1-x)Bi_6Ti_5WO_{22}$–$xBi_6Ti_4Nb_2O_{22}$ ($0 \le x \le 0.6$), with different phase transition temperatures were successfully prepared through ion substitution design. Characterization

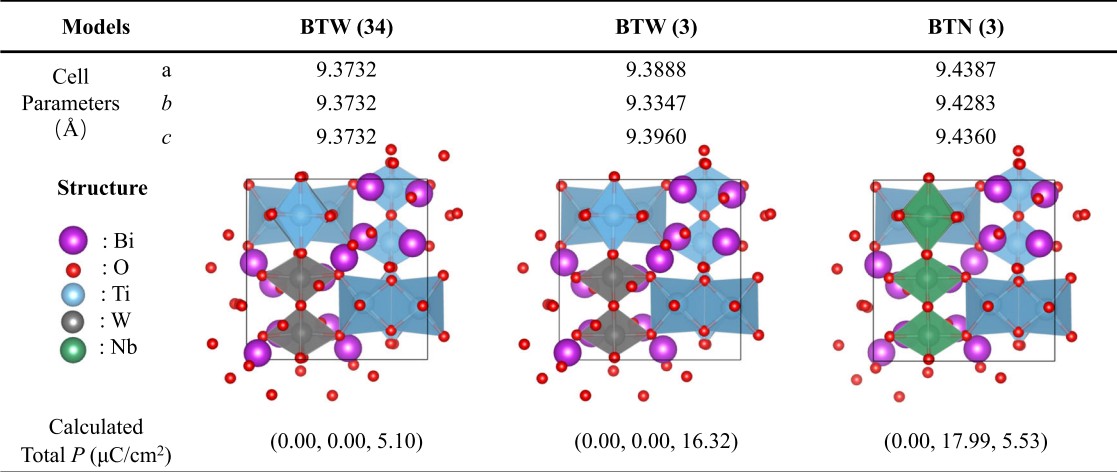

| Models | | BTW (34) | BTW (3) | BTN (3) |
|---|---|---|---|---|
| Cell Parameters (Å) | a | 9.3732 | 9.3888 | 9.4387 |
| | b | 9.3732 | 9.3347 | 9.4283 |
| | c | 9.3732 | 9.3960 | 9.4360 |
| Structure | | | | |
| Calculated Total $P$ ($\mu C/cm^2$) | | (0.00, 0.00, 5.10) | (0.00, 0.00, 16.32) | (0.00, 17.99, 5.53) |

Structure legend:
- : Bi
- : O
- : Ti
- : W
- : Nb

**Fig. 5 | DFT calculation results regarding the introduction of asymmetry and ion substitution.** In the crystal structure diagrams, five differently colored and sized spheres are used to represent five types of ions. The three structural diagrams represent three models optimized using the primitive cell for calculations, namely BTW(34): the cubic phase $Bi_6Ti_5WO_{22}$, BTW(3): the orthorhombic phase $Bi_6Ti_5WO_{22}$, and BTN(3): the orthorhombic phase $Bi_6Ti_4Nb_2WO_{22}$.

of ferroelectric/dielectric properties reveals that through the incorporation of $Nb^{5+}$ with a larger radius, the $T_m$ of BTW ceramics increases, the dielectric constant increases, the dielectric loss is maintained at a very low level, and the dielectric tunability is greatly improved. As a dielectric tunable material, its transcendence in various parameters, including effective tunability, makes it expected to replace the BST solid solution ceramics and promote the wider application of tunable materials in microwave dielectrics and sensor devices.

## Methods
### Sample preparation
Dense lead-free ceramics with the composition of $(1-x)$ $Bi_6Ti_5WO_{22}-xBi_6Ti_4Nb_2O_{22}$ ($0 \leq x \leq 0.6$) were prepared using the conventional high-temperature solid-state reaction method. The raw powder materials included bismuth oxide ($Bi_2O_3$, 99%), tungsten oxide ($WO_3$, 99%), titanium dioxide ($TiO_2$, 99.8%), and niobium oxide ($Nb_2O_5$, 99.9%). Before weighing, all materials underwent initial sintering at 300 °C to remove moisture. Based on process experience, the Bi elements in the composition were weighed at a deficiency of 1.6%. After ball milling in ethanol solvent for 6 hours, the obtained powder was dried and preheated for 4 hours. Subsequently, secondary ball milling was performed under the same conditions. The shaping($\phi$10 mm × 1 mm) process employed a cold isostatic pressing (IP) technique at 200 MPa. The pressed particles were then sintered in the range of 920 °C to 1050 °C for 4 hours, resulting in the final desired ceramic specimens. All the tests and characterizations in our work were conducted based on samples obtained at the optimal sintering temperature (samples of the maximum density). Specifically, the sintering temperature for compositions $x = 0$ and $x = 0.1$ was 1000 °C, for $x = 0.2$ it was 1020 °C, for $x = 0.3$ and $x = 0.4$ it was 1030 °C, and for $x = 0.5$ and $x = 0.6$ it was 1050 °C.

### Structure characterizations
The determination of the phase structure of the material was obtained through X-ray powder diffraction and in-situ neutron diffraction. X-ray diffraction was conducted using a Rigaku D/MAX-2400 with Cu K$\alpha$ radiation and a PXIcel 1D detector, utilizing Cu K$\alpha$1 radiation. The data collection spanned the angular range of $10 \leq 2\theta/° \leq 120$, employing a step size $\Delta 2\theta = 0.008°$. Neutron diffraction at different temperatures from 25 °C to 200 °C was performed using the high-resolution powder diffractometer ECHIDNA at ANSTO. The data collection spanned the angular range of $8 \leq 2\theta/° \leq 160$, employing a step size $\Delta 2\theta = 0.05°$ and a wavelength of 1.622 Å. The combination of PXRD and

NPD data underwent Rietveld refinements using the GSAS program, and the peak shape function developed by van Laar and Yelon was applied[50,51]. Density measurements were determined by sintering large blocks of ceramic samples using the Archimedean displacement method. The relative density was obtained by comparing the measured density with the theoretical density obtained through Rietveld refinement.

### Electron microscopy studies
The microstructure of the ceramics was observed by a field-emission scanning electron microscopy (FE-SEM, FEI Quanta 250 FEG, Hillsboro, Oregon, USA) operating at 30 kV. Before SEM measurements, the sample underwent flat polishing with diamond suspension and was re-calcined at 950 °C for 30 minutes as a hot etching step. The images from the HRTEM (high-resolution transmission electron microscopy) and the HAADF-STEM (high-angle annular dark-field scanning transmission electron microscopy) were obtained using a JEOL F200 and an FEI Cubed Themis G2, respectively. The accelerating voltages used for imaging were 200 kV and 300 kV, respectively.

### Ferroelectric and dielectric measurements
The ferroelectric properties, particularly the polarization-electric field (P-E) loops at different temperatures, were investigated using a probe station equipped with a Precision Premier II Ferroelectric Tester from Radiant Technologies. Before testing, the ceramic samples were meticulously polished to produce thin ceramic layers with a thickness of approximately 0.1 mm. Electrodes were fabricated on the ceramic surface using gold sputtering, with a sputtering time of 2 minutes and an electrode thickness of less than 100 nm. The dielectric spectra over a wide frequency range (from $10^2$ Hz to $10^7$ Hz) and temperature range ($-200$ °C to $+100$ °C) were systematically recorded using an LCR spectrometer (HP4980, Agilent) coupled with a temperature control system (CRYO 610 L). Before testing, low-temperature silver paste was applied to the two polished and smooth surfaces of the sintered sample. The sample was then fired at 55 °C for 30 minutes to create the electrodes. Reversible nonlinear dielectric measurements at a frequency of 10 kHz were conducted using a self-assembled test circuit system. This system comprised a DC voltage stabilizer(SRS350), impedance analyzer(TH2828S), and protective circuit.

### PFM Measurements
The atomic force microscopy tests were conducted using the instrument MFP-3D with a bias voltage module, manufactured by Asylum

Research, USA. The samples used in the experiments were sheet samples, and the surfaces were thoroughly polished and polished before testing.

## DFT calculations

The first-principles calculations are carried out using GPAW[52] with the Perdew-Burke-Ernzerhof (PBE) exchange-correlation functional[53]. To ensure convergence, when performing Brillouin-zone integration on a $4 \times 4 \times 4$ mesh, we set the energy cutoff of 750 eV. Structural relaxations are performed until forces on all atoms are less than 0.05 eV/Å, with an energy convergence criteria set to $5 \times 10^{-4}$ eV. Subsequently, the polarization of each relaxed configuration was calculated using the Berry phase approach[54]. We also calculated the Born effective charge using GPAW, and used it to calculate the polarization as $P = \Sigma d_i \cdot Z_i^* / V$, where $d_i$ is the cation polar displacement, $Z_i^*$ is the Born effective charge of the ions, and $V$ is the volume of the unit cell[55]. These calculations were conducted on $Pnn2$ of $Bi_6Ti_5WO_{22}$ and $Bi_6Ti_4Nb_2O_{22}$, with a pseudo-cubic constraint indicated by $Pn\bar{3}$.

## Data availability

All data supporting the findings are provided as figures and accompanying tables in the article and Supplementary Information. The data that support the findings of this study are available from the corresponding authors.

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

## Acknowledgements

We thank the National Key R&D Program of China (2021YFB3800602) (D. Z.), the National Natural Science Foundation of China (51972260, 52072295, 62175056) (D. Z.), and (52202146) (D. X.), the International Cooperation Project of Shaanxi Province (2021KWZ-10) (D. Z.), the Fundamental Research Funds for the Central University, the 111 Project of China (B14040) (D. Z.). The SEM work was done at the International Center for Dielectric Research (ICDR), Xi'an Jiaotong University, Xi'an, China. The authors thank Dr. Yan-Zhu Dai for her help with SEM.

## Author contributions

R.L. proposed the idea, developed the concept, synthesized all materials, and conducted characterizations. D.X. performed TEM imaging and related analyses. Q.M. and X.L. assisted in AFM testing. F.Z., D.W. and Z.S. conducted simulation and computational modeling work. W.L. assisted in dielectric performance testing. R.L., C.D. and D.Z. were the primary writers of the paper and the principal investigators. All authors discussed the results and provided feedback on the paper.

## Competing interests

The authors declare no competing interests.
