## [Peer Review File · Nature Communications]

Giant dielectric tunability in ferroelectric ceramics with ultralow loss by ion substitution designREVIEWER COMMENTS

Reviewer #1 (Remarks to the Author):

attached

Reviewer #2 (Remarks to the Author):

Manuscript "Giant dielectric tunability in ferroelectric ceramics with ultralow loss by ion substitution design" reports giant tunability (~75.6%) with ultralow loss (< 0.002) under a low electric field (1.5 kV/mm). Authors have reached on the conclusion that "the incorporation of Nb⁵⁺ with a larger radius, the T_m of BTW ceramics increases, the dielectric constant increases, the dielectric loss is maintained at a very low level, and the dielectric tunability is greatly improved". Manuscript is well written and observations are supported with good quality experimental and theoretical studies. Reported results are the classic example of the physical property change due to the interactions among different valence state of atoms in chemically modified compositions.

Manuscript can be accepted after taking care of following observations:

- On page 2, Bi₆Ti₅WO₂₂ should be Bi₆Ti₅WO₂₂
- Fig1, XRD patterns should be indexed.
- Thin plate shaped secondary phase is found due to the increase in Nb content. EDS data is showing the presence of Nb in it (supplementary data), why have authors not reported Nb content in impurity phase?
- How did author identified impurity composition Bi₆Ti₃WO₁₈? It should be included in manuscript. References can be added.
- How did authors manage to quantify the Bi content in final compositions? It should be included for clarity.
- Figure 2d does not show the graph of five compositions, more clearer graph can be included.
- Figure 4e, if authors can show the effect of the presence of Nb in lattice, it will give clearer view to readers.
- Authors may include more recent years references related to low dielectric loss in other related systems, which may attract more readers.

Reviewer #3 (Remarks to the Author):

This article entitled "Giant dielectric tunability in ferroelectric ceramics with ultralow loss by ion substitution design" focuses on achieving "giant" dielectric tunability with "ultralow" loss by incorporating Nb ions in the solid solutions of $(1-x)\text{Bi}_6\text{Ti}_5\text{WO}_{22} - x\text{Bi}_6\text{Ti}_4\text{Nb}_2\text{O}_{22}$ ($0 \leq x \leq 0.6$). The authors presented several intriguing and well-executed experimental and computational findings. Nevertheless, the following points should be clarified or strengthened before publication:

General:

1. The prime idea of this work is to incorporate Nb ions into the ferroelectric $(1-x)\text{Bi}_6\text{Ti}_5\text{WO}_{22} - x\text{Bi}_6\text{Ti}_4\text{Nb}_2\text{O}_{22}$ to enhance the dielectric tunability while minimizing loss. However, there remains a lack of direct evidence confirming the ion substitution. The authors used energy dispersive spectroscopy (EDS) to verify the existence of Nb ions in the ferroelectric solid solutions. Nevertheless, the obtained Bi:Ti:W:Nb atomic ratios presented in Supplementary Fig. 1 deviate significantly from the expected values. This discrepancy may

be attributed to the limited resolution of EDS and the overlapping characteristic peaks of Ti-L α (= 0.452 keV) and O-K α (= 0.525 keV) in the EDS spectra. It is recommended to employ more suitable techniques, such as X-ray photoelectron spectroscopy (XPS) or electron energy loss spectroscopy (EELS) for more precise analyses.

2. The effective tunability (T_0) has been denoted inconsistently within the manuscript; the authors referred to it as T_e in the definition given in Eq. (2) but used T_0 in the Fig. 3b. It is defined as T/E , where T represents the dielectric tunability at a specific applied electric bias E ($E \geq 0.5$ kV/mm), as per Ref. [7]. As T_0 represents the slope of tunability, it is recommended to plot T vs. E curves and explicitly state the applied electric bias used to compute the effective tunability.

In the Abstract, it is noted that in the $(1-x)\text{Bi}_6\text{Ti}_5\text{WO}_{22} - x\text{Bi}_6\text{Ti}_4\text{Nb}_2\text{O}_{22}$ solid solutions, when $x = 0.3$, the reported tunability was approximately 75.6% (or $T \sim 76.7\%$ as seen in Supplementary Fig. 9b; both values are inconsistent) under an electric field of 1.5 kV/mm. Consequently, the effective tunability should be calculated as 0.504 mm/kV (or 0.511 mm/kV), which contrasts with the value of 0.634 mm/kV provided in the text. Similarly, for $x = 0.4$, where the reported tunability is approximately 87.5% (as indicated in Supplementary Fig. 9c under 1.5 kV/mm?), the effective tunability should be 0.584 mm/kV instead of the claimed value of 0.805 mm/kV. These values should be verified for accuracy.

3. The figure of merit should be accurately defined. In the authors' previous publication (Ref. [32]), they defined it as "the ratio of dielectric loss to tunability," which is incorrect. Instead, the figure of merit should be defined as the reciprocal of this ratio, I believe, the ratio of tunability to dielectric loss.

4. During Rietveld refinement, the "occupancy" is one of the parameters adjusted to improve the agreement between the calculated diffraction pattern and the experimental data. In the case of $0.6\text{Bi}_6\text{Ti}_5\text{WO}_{22} - 0.4\text{Bi}_6\text{Ti}_4\text{Nb}_2\text{O}_{22}$ ($x = 0.4$), as detailed in Supplementary Tables 1-4, the total occupancy of Nb is specified as 0.4. However, discrepancies arise concerning the total occupancy of W, which is approximately 0.3215, and total occupancy for Ti, recorded as 2.2785 at 25, 50, and 100°C, but reduced to 1.824 at 200°C. These all deviate from their expected stoichiometric ratios. Please recheck the results.

5. During the DFT calculations, it is recommended to report the convergence level.

Typos:

There are several typos that need to be corrected, to name a few:

1. (line 79) " $\text{Ba}_6\text{Ti}_5\text{O}_2$ " should be " $\text{Ba}_6\text{Ti}_5\text{O}_{22}$ "
2. (line 106) "Fig. 1c" should be "Fig. 1e"
3. (line 112) XRD peak shifts to "higher" angles should be "lower"
4. (line 237) "Fig. 4a" should be "Fig. 4a-4c"
5. (line 239) "Fig. 4b" should be "Fig. 4d"
6. (line 268) "repla" should be "replaces"

The authors have successfully synthesized Nb⁵⁺ modified Bi₆Ti₅WO₂₂ ceramics by solid state reaction method. The authors have prepared a series of (1-x)Bi₆Ti₅WO₂₂-xBi₆Ti₄Nb₂O₂₂ ceramics with x = 0, 0.10, 0.20, 0.30, 0.40, 0.50 and 0.60. Authors have characterized the prepared samples with XRD, SEM, EDS, dielectric and ferroelectric properties along with neutron powder diffraction, TEM and DFT for in depth analysis to understand the polarization behaviour. Authors focused on the polarization behaviour under external DC field. The results are technically sounds and enough to justify the enhanced dielectric tunability for the prepared ceramics by Nb substitution. But there are few below points where authors can improve the manuscript.

1. In sentence, "Considering that a larger lattice constant aids in achieving a higher T_m, introducing Nb⁵⁺ with an ionic radius larger than both Ti⁴⁺ 90 (0.64 Å compared to 0.605 Å) and W⁶⁺ (0.64 Å compared to 0.60 Å) is a suitable choice", authors need to justify why Nb⁵⁺ is the preferable choice for substitution. In overall, authors need to justify the choice of Nb⁵⁺ ion.
2. The grain size increased with Nb content. Why?
3. In sentence "Moreover, the refined results of the room-temperature XRD patterns indicate that, with the increasing Nb content, the main XRD peak shifts to higher angles, indicating an enlargement of lattice constants due to the successful introduction of Nb (as depicted in Fig. 1d)". However, the peaks are shifting towards lower angle. The sentence need to be reframed.
4. Supplementary fig 2 does not have refined XRD pattern. How the lattice parameters were calculated?

Therefore, the manuscript can be accepted after a minor revision by improving the manuscript on the above points.

Dear Editor and Reviewers:

Thanks a lot for your letter and for the reviewers' comments concerning our manuscript entitled "**Giant dielectric tunability in ferroelectric ceramics with ultralow loss by ion substitution design**" (Manuscript Number: NCOMMS-24-03584-T). Those comments are all valuable and very helpful for revising and improving our paper, as well as the important guiding significance to our research. We have studied the comments carefully and have made the following corrections which we hope to meet with approval. All the corrections are marked in red through the text. The responses to the reviewers' comments are listed as follows:

Reply to Reviewer # 1:

The authors have successfully synthesized Nb⁵⁺ modified Bi₆Ti₅WO₂₂ ceramics by solid state reaction method. The authors have prepared a series of (1-x)Bi₆Ti₅WO₂₂ - xBi₆Ti₄Nb₂O₂₂ ceramics with $x = 0, 0.10, 0.20, 0.30, 0.40, 0.50$ and 0.60 . Authors have characterized the prepared samples with XRD, SEM, EDS, dielectric and ferroelectric properties along with neutron powder diffraction, TEM and DFT for in depth analysis to understand the polarization behavior. Authors focused on the polarization behavior under external DC field. The results are technically sounds and enough to justify the enhanced dielectric tunability for the prepared ceramics by Nb substitution. But there are few below points where authors can improve the manuscript.Therefore, the manuscript can be accepted after a minor revision by improving the manuscript on the above points.

Reply: We sincerely thank reviewer #1 for your positive comments and for recommending our manuscript for publication in *Nature Communications*. Point-by-point responses to your comments are listed below.

1. In sentence, "Considering that a larger lattice constant aids in achieving a higher T_m, introducing Nb⁵⁺ with an ionic radius larger than both Ti⁴⁺ (0.64 Å compared to 0.605 Å) and W⁶⁺ (0.64 Å compared to 0.60 Å) is a suitable choice", authors need to justify why Nb⁵⁺ is the preferable choice for substitution. In overall, the authors need to justify the choice of Nb⁵⁺ ion.

Reply: Thank you for your comment. It has been demonstrated in many perovskite systems that forming solid solutions by doping elements with different ionic radii can adjust the material's phase transition temperature. ^[1] In previous work, we determined the precise crystal structure of $\text{Bi}_6\text{Ti}_5\text{WO}_{22}$, confirmed the specific situation of Ti/W mixed occupancy, and believed that ions at this position significantly contribute to polarization. ^[2] Therefore, we believe that ion substitution at this position can effectively regulate the material's properties, which is the starting point of this work.

As for the choice of Nb^{5+} ions, there are mainly three reasons:

a. The similar ion radius allows Nb^{5+} ions to bind to the existing lattice framework to form a solid solution without generating a second phase in limited compositions;

b. The slightly larger ion radius enables the introduction of Nb^{5+} ions to induce a change in lattice constant, thereby adjusting the phase transition temperature and affecting the dielectric properties;

c. The pentavalent state of Nb^{5+} ions ensure the structural charge balance is maintained when replacing the Ti/W ions at the shared position.

These three points make Nb^{5+} the best choice for our ion substitution design, which has been proven effective in subsequent experiments. We have revised the relevant descriptions in the main text to make our ideas clearer to the readers.

[1] Liu, X., Yin, J. & Wu, J. A new class of ion substitution to achieve high electrostrain under low electric field in BNT-based ceramics. *Journal of the American Ceramic Society* **104**, 6277-6289 (2021).

[2] Li, R. et al. Ultralow loss and high tunability in a non-perovskite relaxor ferroelectric. *Advanced Functional Materials* **33**, 2210709 (2023).

2. The grain size increased with Nb content. Why?

Reply: Thank you for pointing out the issue. The grain size is generally determined by the sintering temperature and the sintering process.

All the tests and characterizations in our work were conducted based on samples obtained at the optimal sintering temperature (samples of the maximum density). Since the introduction of Nb slightly increases the sintering temperature of the ceramic samples, the sintering temperatures of the samples shown in the SEM image in Fig. 1e are not the same. Specifically, the sintering temperature for compositions $x = 0$ and $x = 0.1$ was 1000 °C, for $x = 0.2$ it was 1020 °C, for $x = 0.3$ and $x = 0.4$

it was 1030 °C, and for $x = 0.5$ and $x = 0.6$ it was 1050 °C. Different sintering temperatures result in an increase in grain size observed with the introduction of Nb.

The influence of grain size variation at the micrometer scale on performance is generally not significant. Our previous experience has also verified that the variation in grain size of this series of solid solutions has no effect on dielectric performance, especially the phase transition temperature. Therefore, the influence of grain size on performance was not additionally considered in the subsequent performance analysis.

We have added an explanation for the increase in grain size in the main text, and a statement regarding the different optimal sintering temperatures has been added to the experimental methods section of the manuscript.

3. In sentence “Moreover, the refined results of the room-temperature XRD patterns indicate that, with the increasing Nb content, the main XRD peak shifts to higher angles, indicating an enlargement of lattice constants due to the successful introduction of Nb (as depicted in Fig. 1d)”. However, the peaks are shifting towards lower angle. The sentence needs to be reframed.

Reply: Thank you very much for your carefulness. We made a mistake in the description in the main text and it was corrected. Subsequent analyses were given based on the XRD shift towards low angles. We apologize for this mistake and have corrected all formatting and writing mistakes.

4. Supplementary Fig. 2 does not have refined XRD pattern. How the lattice parameters were calculated?

Reply: Thank you for your reminder. The lattice constants were obtained from refinements of the X-ray diffraction (XRD) data. We have supplemented all the refinement results in the Supplementary Fig. 2.

Reply to Reviewer # 2:

Manuscript “Giant dielectric tunability in ferroelectric ceramics with ultralow loss by ion substitution design” reports giant tunability (~75.6%) with ultralow loss (< 0.002) under a low electric field (1.5 kV/mm). Authors have reached on the conclusion that “the incorporation of Nb⁵⁺ with a larger radius, the T_m of BTW ceramics increases, the dielectric constant increases, the dielectric loss is maintained at a very low level, and the dielectric tunability is greatly improved”.

Manuscript is well written and observations are supported with good quality experimental and theoretical studies. Reported results are the classic example of the physical property change due to the interactions among different valence state of atoms in chemically modified compositions. The manuscript can be accepted after taking care of following observations:

Reply: We greatly appreciate reviewer # 2 for your time to review our manuscript and give us the above extremely encouraging comments. Point-by-point responses to your comments are listed below.

1. On page 2, $\text{Bi}_6\text{Ti}_5\text{WO}_2$ should be $\text{Bi}_6\text{Ti}_5\text{WO}_{22}$.

Reply: Thank you very much for your patience and thorough review of our manuscript. We apologize for any formatting errors or writing mistakes that appeared in the manuscript. We have now carefully and comprehensively corrected them.

2. Fig. 1, XRD patterns should be indexed.

Reply: Thank you for your good suggestions. In the main text, we primarily aimed to describe the structural changes and the shift in diffraction peak positions, so we did not index individual peaks. We have added the XRD identification, indexing, and refinement results in detail in Supplementary Fig. 2.

3. Thin plate shaped secondary phase is found due to the increase in Nb content. EDS data is showing the presence of Nb in it (supplementary data), why have authors not reported Nb content in impurity phase?

Reply: Fig. 1d, Fig. 1e, and Supplementary Fig. 1 are intended to illustrate that the maximum solid solubility of the $(1-x)\text{Bi}_6\text{Ti}_5\text{WO}_{22} - x\text{Bi}_6\text{Ti}_4\text{Nb}_2\text{O}_{22}$ solid solution system is approximately between 40% and 50%. The results of XRD patterns and SEM micrographs together confirm that when $x < 0.5$, the introduction of Nb ions does not produce a second phase or impurities, but when $x > 0.5$, it changes the original lattice structure to produce impurity phases. Subsequent work focuses on solid solution ceramics with $x < 0.5$, and detailed characterization and testing of samples with $x > 0.5$ have not been carried out, so not much effort has been devoted to the analysis of impurities in these samples.

The identification of impurity phases is partly based on the determination of characteristic peaks in XRD patterns. In the XRD diffraction pattern of the $x = 0.6$ ceramic powder, we found characteristic peaks of $\text{Bi}_6\text{Ti}_3\text{WO}_{18}$ (as shown in Fig. 1d).^[1] Additionally, the thin plate-shaped

lamellar structures in the SEM micrographs also match the grain characteristics of the $\text{Bi}_6\text{Ti}_3\text{WO}_{18}$ phase. $\text{Bi}_6\text{Ti}_3\text{WO}_{18}$ is a structure that is easily formed in the Bi-Ti-W ternary system and has often appeared in our previous explorations of pure BTW preparation.^[2] The elemental analysis in Supplementary Fig. 1 is mainly aimed at reaffirming the possible presence of $\text{Bi}_6\text{Ti}_3\text{WO}_{18}$ through a comparison of Bi element contents.

Regarding the issue of Nb content you pointed out, on the one hand, it may be due to errors in EDS scan patterns. The EDS scans a range rather than an exact point, which may explain the presence of Nb elements in the analysis of the sheet samples. We have revised the illustrations in the supplementary information to avoid misunderstandings. On the other hand, through our reexamination of the literature, we have indeed found the $\text{Bi}_3\text{TiNbO}_9$ phase, which has the same structure as $\text{Bi}_6\text{Ti}_3\text{WO}_{18}$. Given the excessive introduction of Nb ions, the impurities should also contain $\text{Bi}_3\text{TiNbO}_9$.^[3] More precise impurity composition determination may be achieved through more sophisticated analytical methods, but this deviates from the focus of our current work.

We greatly appreciate your valuable comments and have added descriptions of the possible presence of $\text{Bi}_3\text{TiNbO}_9$ in the manuscript and Supplementary Fig. 1.

[1] Jardiel T, Villegas M, Caballero A, Suvorov D, Caballero AC. Solid-State compatibility in the system $\text{Bi}_2\text{O}_3\text{-TiO}_2\text{-Bi}_2\text{WO}_6$. *Journal of the American Ceramic Society* **91**, 278-282 (2008).

[2] Li, R. et al. Ultralow loss and high tunability in a non-perovskite relaxor ferroelectric. *Advanced Functional Materials* **33**, 2210709 (2023).

[3] Kikuchi T. Synthesis of new, layered bismuth titanates, $\text{Bi}_7\text{Ti}_4\text{NbO}_{21}$ and $\text{Bi}_6\text{Ti}_3\text{WO}_{18}$. *Journal of the Less Common Metals* **48**, 319-323 (1976).

4. How did author identified impurity composition $\text{Bi}_6\text{Ti}_3\text{WO}_{18}$? It should be included in manuscript. References can be added.

Reply: We appreciate your insightful recommendation. As previously mentioned, the identification of impurity composition is based on the determination of characteristic peaks in XRD patterns, recognition of grain characteristics in SEM micrographs, and elemental analysis from EDS scans. Based on current analysis we preliminarily infer that the impurity phases are probable to be $\text{Bi}_6\text{Ti}_3\text{WO}_{18}$ and $\text{Bi}_3\text{TiNbO}_9$, which are considered to be iso-structural. We have added explanations and references in the manuscript.

5. How did authors manage to quantify the Bi content in final compositions? It should be

included for clarity

Reply: Thank you for pointing out this issue. It has indeed been a challenge for us. We attempted various methods, such as inductively coupled plasma mass spectrometry (ICP-MS), to determine the exact elemental composition of the final samples. However, due to the strong chemical stability of the ceramic we prepared, its powder could not be dissolved in various acid and alkali solutions, including aqua regia, which prevented us from obtaining the corresponding solution for accurate ICP-MS analysis.

Nevertheless, we believe that the $(1-x)\text{Bi}_6\text{Ti}_5\text{WO}_{22} - x\text{Bi}_6\text{Ti}_4\text{Nb}_2\text{O}_{22}$ ($x \leq 0.5$) ceramic samples after ion substitution show no evidence of a second phase in their X-ray diffraction patterns and SEM micrographs, indicating that the final samples are structurally consistent with the undoped ones. The theoretical data for $\text{Bi}_6\text{Ti}_5\text{WO}_{22}$ calculated with a Bi:Ti/W/Nb ratio of 1:1 fit well with the measured powder data from our X-ray and neutron powder diffraction refinements. The proportions obtained from SEM-EDS elemental analysis are also close to this ratio. Therefore, we are confident that the elemental ratio of Bi:Ti/W/Nb = 1:1 is accurate.

6. Figure 2d does not show the graph of five compositions, more clearer graph can be included.

Reply: The data for all five compositions are shown in the figure, but the data for $x = 0.3$ and $x = 0.5$ overlap significantly, making it difficult to distinguish. We apologize for any misunderstanding this may have caused. Given the difficulty in distinguishing them in the same figure, we have explained in the main text and separately plotted P-E curves for each composition in Supplementary Fig. 12, with a total of five figures.

7. Figure 4e, if authors can show the effect of the presence of Nb in lattice, it will give clearer view to readers.

Reply: In Fig. 4e, we aim to illustrate the presence and dimensions of the twin structure and nanoscale domains in BTW-BTN ceramics. Readers can compare our previously published TEM images of pure BTW ceramics to understand the effects of Nb ion substitution.^[1] We have added descriptions of the changes in micrographs of BTW-BTN ceramics compared to pure BTW ceramics in our manuscript, and have included some comparative images in the supplementary information. It can be observed that due to the introduction of Nb ions, the nanodomains are denser, clearer, and smaller.

Of course, we are also very interested in using microscopic observations to investigate the

specific arrangement of Nb ions in the lattice and nanoscale polar regions, as well as their specific effects on the lattice structure. However, the results from EDX for these very small nanodomains were poor, and we were unable to obtain ideal images. This will be the focus of our next research step.

[1] Li, R. et al. Ultralow loss and high tunability in a non-perovskite relaxor ferroelectric. *Advanced Functional Materials* **33**, 2210709 (2023).

8. Authors may include more recent years references related to low dielectric loss in other related systems, which may attract more readers.

Reply: We appreciate your insightful recommendation. We have added the following references to the list of citations.

[1] Gu Z, et al. Resonant domain-wall-enhanced tunable microwave ferroelectrics. *Nature* **560**, 622-627 (2018).

[2] Lee C, et al. Exploiting dimensionality and defect mitigation to create tunable microwave dielectrics. *Nature* **502**, 532-536 (2013).

[3] Guo Y, et al. Characterization and performance of plate-like Ba_{0.6}Sr_{0.4}TiO₃/Poly (vinylidene fluoride–trifluoroethylene-chlorotrifluoroethylene) composites with high permittivity and low loss. *Polymer* **203**, 122777 (2020).

Reply to Reviewer # 3:

This article entitled "Giant dielectric tunability in ferroelectric ceramics with ultralow loss by ion substitution design" focuses on achieving "giant" dielectric tunability with "ultralow" loss by incorporating Nb ions in the solid solutions of $(1-x)\text{Bi}_6\text{Ti}_5\text{WO}_{22} - x\text{Bi}_6\text{Ti}_4\text{Nb}_2\text{O}_{22}$ ($0 \leq x \leq 0.6$). The authors presented several intriguing and well-executed experimental and computational findings. Nevertheless, the following points should be clarified or strengthened before publication:

Reply: We greatly appreciate the Reviewer # 3 to review our manuscript. The enlightening suggestions are very helpful for improving this work. Thus, we have revised our manuscript accordingly. Point-by-point responses to the reviewers' comments are also listed below.

1. The prime idea of this work is to incorporate Nb ions into the ferroelectric $(1-x)\text{Bi}_6\text{Ti}_5\text{WO}_{22} - x\text{Bi}_6\text{Ti}_4\text{Nb}_2\text{O}_{22}$ to enhance the dielectric tunability while minimizing loss. However, there remains

a lack of direct evidence confirming the ion substitution. The authors used energy dispersive spectroscopy (EDS) to verify the existence of Nb ions in the ferroelectric solid solutions. Nevertheless, the obtained Bi:Ti:W:Nb atomic ratios presented in Supplementary Fig. 1 deviate significantly from the expected values. This discrepancy may be attributed to the limited resolution of EDS and the overlapping characteristic peaks of Ti-L α (= 0.452 keV) and O-K α (= 0.525 keV) in the EDS spectra. It is recommended to employ more suitable techniques, such as X-ray photoelectron spectroscopy (XPS) or electron energy loss spectroscopy (EELS) for more precise analyses.

Reply: Thank you for pointing out the issue. However, as a rough estimation of elemental proportions, the results obtained from EDS analysis are generally consistent with our expectations:

- a. The ratio of Bi to Ti/W/Nb is consistently close to 1:1.
- b. With an increase in Nb content, there is a corresponding decrease in Ti/W content.
- c. The measured proportion of Nb elements matches the actual number of Nb ions introduced, with an error within 15%, which is attributed to the limited resolution of EDS and the overlapping characteristic peaks in the EDS spectra as you mentioned.

We believe this is sufficient to qualitatively demonstrate the effective substitution of Nb.

Additionally, for the ceramic samples of $(1-x)\text{Bi}_6\text{Ti}_5\text{WO}_{22} - x\text{Bi}_6\text{Ti}_4\text{Nb}_2\text{O}_{22}$ ($x \leq 0.5$), no second phase was observed in either the X-ray diffraction or SEM micrographs. This suggests that no unexpected products were formed beyond the intended raw materials. Subsequent fitting of X-ray powder diffraction and neutron powder diffraction refinement data also supports the gradual increase in Nb content in the final samples with increasing Nb introduced in the initial powder. We have reason to believe that ion substitution proceeded as we anticipated.

As for the more precise elemental proportions in the samples, we attempted various methods, including inductively coupled plasma mass spectrometry (ICP-MS), to determine the exact elemental composition of the final samples. However, due to the strong chemical stability of the ceramic we prepared, its powder could not be dissolved in various acid and alkali solutions, including aqua regia, which prevented us from obtaining the corresponding solution for highly accurate measurement.

2. The effective tunability (T_0) has been denoted inconsistently within the manuscript; the authors referred to it as T_e in the definition given in Eq. (2) but used T_0 in the Fig. 3b. It is defined

as T/E , where T represents the dielectric tunability at a specific applied electric bias E ($E \geq 0.5$ kV/mm), as per Ref. [7]. As T_0 represents the slope of tunability, it is recommended to plot T vs. E curves and explicitly state the applied electric bias used to compute the effective tunability. In the Abstract, it is noted that in the $(1-x)\text{Bi}_6\text{Ti}_5\text{WO}_{22} - x\text{Bi}_6\text{Ti}_4\text{Nb}_2\text{O}_{22}$ solid solutions, when $x = 0.3$, the reported tunability was approximately 75.6% (or $T \sim 76.7\%$ as seen in Supplementary Fig. 9b; both values are inconsistent) under an electric field of 1.5 kV/mm. Consequently, the effective tunability should be calculated as 0.504 mm/kV (or 0.511 mm/kV), which contrasts with the value of 0.634 mm/kV provided in the text. Similarly, for $x = 0.4$, where the reported tunability is approximately 87.5% (as indicated in Supplementary Fig. 9c under 1.5 kV/mm?), the effective tunability should be 0.584 mm/kV instead of the claimed value of 0.805 mm/kV. These values should be verified for accuracy.

Reply: Thank you for your review of the data consistency and your comments on the effective tunability of our manuscript.

The difference in tunability at 1.5 kV/mm between the abstract and Supplementary Fig. 9 arises from a computational error in the calculation method. As shown in the supplementary material, tunability is tested in a complete cycle from 0 to positive voltage, then to negative voltage, and back to 0 bias. In the calculation of tunability in Supplementary Fig. 9, the dielectric constant at 0 electric field and the lowest dielectric constant after applying positive and negative voltages were used, resulting in the maximum tunability calculated.

In the abstract, we used the mean tunability during a single process from multiple cycles, which results in a slightly lower tunability value but is more scientifically sound and conservative. We have now revised all tunability values in Supplementary Fig. 9 to the mean tunability values.

The concept of effective tunability was first proposed by Hu to compare the dependence of BST-based dielectric tunable materials on the bias electric field.^[1] In this work, to distinguish it from temperature symbols, we define effective tunability as T_e . (The mistake in Fig. 3b has been corrected.)

For most materials, the dielectric tunability is not linear with the change of the applied electric field. This results in the value of T_e changing with different applied bias field intensities. The discrepancy in the data you mentioned is due to the different applied electric field intensities selected for calculating the effective tunability T_e .

Reintroducing the concept of T_e and plotting Fig. 3b, we aim to highlight a key advantage of BTW-BTN materials over other ferroelectric ceramics: the ability to achieve giant dielectric tunability under a low applied electric field. In Fig. 3b, we selected an applied electric field intensities of 1 kV/mm to calculate T_e . For 0.6BTW-0.4BTN ceramics, the T_e remains above 0.8 mm/kV in the range of 0.3 to 1.1 kV/mm, and for 0.7BTW-0.3BTN ceramics, the T_e remains above 0.6 mm/kV in the range of 0.5 to 1.2 kV/mm. We believe that these two values reported at 1 kV/mm objectively reflect the low-field tunability of BTW-BTN materials. (For 0.6BTW-0.4BTN ceramics, the value of T_e will be as high as 1.14 mm/kV if calculated at 0.5 kV/mm; for 0.7BTW-0.3BTN ceramics, the value of T_e will be 0.676 mm/kV if calculated at 0.7 kV/mm. We did not intentionally select higher values.)

Following your advice, we have supplemented the explanation of the applied electric field selected for calculating T_e in the manuscript and modified the data of the comparative materials in Fig. 3b (selecting the values of T_e also calculated under 1 kV/mm), making the data more rigorous.

Overall, there are many factors to consider in the practical application of dielectric tunable materials: dielectric constant, loss, tunability, the applied electric field required to achieve the tunability, test frequency, response time, etc., all of which are crucial for achieving effective phase shifting.^[2] Currently, there is no comprehensive index to fully evaluate the merits of materials. In practical research comparisons, it is easy to only compare a single parameter while ignoring others. Therefore, we have also included Fig. 3d in our manuscript: This is a comparison of BTW-BTN materials with BST system materials of similar dielectric constants under the same applied electric field intensity, same measurement frequencies, the same temperature, and the same testing methods after being normalized. This comparison can scientifically and comprehensively reflect the advantage of giant tunability under low electric fields of BTW-BTN materials.

[1] Hu, G., Gao, F., Liu, L., Xu, B. & Liu, Z. Microstructure and dielectric properties of highly tunable Ba_{0.6}Sr_{0.4}TiO₃/MgO/Al₂O₃/ZnO composite. *Journal of Alloys and Compounds* **518**, 44-50 (2012).

[2] Gao, F., Zhang, K., Guo, Y., Xu, J. & Szafran, M. (Ba, Sr) TiO₃/polymer dielectric composites—progress and perspective. *Progress in Materials Science* **121**, 100813 (2021).

3. The figure of merit should be accurately defined. In the authors' previous publication (Ref. [32]), they defined it as "the ratio of dielectric loss to tunability," which is incorrect. Instead, the

figure of merit should be defined as the reciprocal of this ratio, I believe, the ratio of tunability to dielectric loss.

Reply: We appreciate your insightful recommendation. As you mentioned, the figure of merit is the ratio of tunability to dielectric loss, which is a consensus understanding in the field of tunable dielectrics. We have supplemented its definition in the main text: “Moreover, the figure of merit (FOM) is also a common parameter used to evaluate the quality of dielectric tunable materials and is defined as the ratio of tunability to dielectric loss.”

4. During Rietveld refinement, the “occupancy” is one of the parameters adjusted to improve the agreement between the calculated diffraction pattern and the experimental data. In the case of $0.6\text{Bi}_6\text{Ti}_5\text{WO}_{22} - 0.4\text{Bi}_6\text{Ti}_4\text{Nb}_2\text{O}_{22}$ ($x = 0.4$), as detailed in Supplementary Tables 1-4, the total occupancy of Nb is specified as 0.4. However, discrepancies arise concerning the total occupancy of W, which is approximately 0.3215, and total occupancy for Ti, recorded as 2.2785 at 25, 50, and 100 °C, but reduced to 1.824 at 200 °C. These all deviate from their expected stoichiometric ratios. Please recheck the results.

Reply: As you mentioned, the occupancy rate is a one of the parameters adjusted to improve the agreement between the calculated diffraction pattern and the experimental data. During the refinement of our composition, the occupancy rate of Nb was specified as 0.4. The mention of 200 °C as an anomaly in the Ti data was a mistake in the transcription process. We apologize for any misunderstanding caused by this oversight. We have made corrections in the manuscript and carefully reviewed the refinement data and other data in the manuscript.

5. During the DFT calculations, it is recommended to report the convergence level.

Reply: We are grateful to the reviewers for pointing out the issue. In our calculations, we utilized the default convergence settings of GPAW, with an energy convergence criterion of 5×10^{-4} eV. In practice, the final variations in total energy were all within 1×10^{-5} eV. We described the relevant arguments in the DFT calculations section of our Methods. We have added a more detailed description regarding energy convergence in this part.

Typos: There are several typos that need to be corrected, to name a few: **1.** (line 79) " $\text{Bi}_6\text{Ti}_5\text{WO}_2$ " should be " $\text{Bi}_6\text{Ti}_5\text{WO}_{22}$ ". **2.** (line 106) "Fig. 1c" should be "Fig. 1e". **3.** (line 112) XRD peak shifts to "higher" angles should be "lower". **4.** (line 237) "Fig. 4a" should be "Fig. 4a-4c". **5.** (line 239) "Fig. 4b" should be "Fig. 4d". **6.** (line 268) "repla" should be "replaces".

Reply: Thank you very much for your patience and thorough review of our manuscript. We apologize for any formatting errors or typographical mistakes that appeared in the manuscript. We have now carefully and comprehensively corrected them.

Sincerely yours,

Prof. Dr. Di Zhou

Xi'an Jiaotong University

13-Mar.-2024

REVIEWERS' COMMENTS

Reviewer #1 (Remarks to the Author):

Authors have now revised the manuscript and clarify all the queries. Therefore, the manuscript can be accepted for publication.

Reviewer #2 (Remarks to the Author):

Authors have incorporated all the corrections in the revised manuscript except indexing of XRD plots. I am satisfied with the corrections and discussions, which are included in this revised manuscript. In my view, Revised manuscript is suitable for publication in its present form.

Reviewer #3 (Remarks to the Author):

Most of my previous comments have been adequately addressed.

Reviewers' comments:

Reviewer #1 (Remarks to the Author):

Authors have now revised the manuscript and clarify all the queries. Therefore, the manuscript can be accepted for publication.

Reply: We express our sincere gratitude for your thorough review of our manuscript. Your insightful feedback and constructive comments have been immensely valuable in refining our work.

We have diligently addressed each of your concerns and incorporated your suggested revisions. We believe that these changes have significantly enhanced the clarity, rigor, and overall quality of the research presented.

Once again, we sincerely thank you for your positive comments and for recommending our manuscript for publication in *Nature Communications*. We extend our appreciation for your time and expertise in evaluating our work. Your efforts have undoubtedly played a crucial role in improving the scholarly merit of our paper.

Reviewer #2 (Remarks to the Author):

Authors have incorporated all the corrections in the revised manuscript except indexing of XRD plots. I am satisfied with the corrections and discussions, which are included in this revised manuscript. In my view, Revised manuscript is suitable for publication in its present form.

Reply: We sincerely appreciate your comprehensive review of our manuscript. The constructive suggestions and insights you provided have clearly helped us improve the rigor and clarity of the manuscript. Your expertise has been invaluable in shaping the final version. We also appreciate your recognition of the corrections and modifications we made to the manuscript.

Regarding the indexing of XRD plots, considering that Fig.1d is primarily used to demonstrate the generation of a secondary phase and peak shifts due to Nb doping, we believe that the current labeling adequately explains the issues. Providing additional crystallographic information would affect the clarity and aesthetics of the figure. Therefore, after careful consideration, we have decided not to add further annotations to Fig.1d. We believe that readers seeking more detailed information about the crystals can refer to the Supplementary Fig. 2.

Once again, thank you for your invaluable contribution to this work.

Reviewer #3 (Remarks to the Author):

Most of my previous comments have been adequately addressed.

Reply: Thank you for reviewing our manuscript once again, and for acknowledging that we have addressed your comments. We appreciate your meticulous evaluation and valuable feedback, which has undoubtedly contributed to enhancing the quality of our work.

We have taken each of your suggestions seriously and carefully implemented the necessary revisions to ensure a more comprehensive and coherent presentation of our findings. Your expertise and critical insights have been instrumental in shaping the final version of the manuscript.

Considering the thorough revisions made, we believe that the manuscript now meets the rigorous standards of the journal.

Once again, thank you for your invaluable contribution to this work.